# Analysis of the Bovine DLK1 Gene Polymorphism and Its Relation to Lipid Metabolism in Chinese Simmental

**DOI:** 10.3390/ani10060923

**Published:** 2020-05-26

**Authors:** Mengyan Wang, Ping Jiang, Xiang Yu, Jiaqi Mi, Zitong Bai, Xiuqi Zhang, Yinuo Liu, Xibi Fang, Runjun Yang, Zhihui Zhao

**Affiliations:** 1College of Agriculture, Guangdong Ocean University, Zhanjiang 524088, China; wangmengyan17@163.com (W.M.); 15584169529@163.com (P.J.); 2College of Animal Science, Jilin University, Changchun 130062, China; fisheryx@163.com (X.Y.); jlumijiaqi@163.com (J.M.); baizitong96@163.com (Z.B.); zxq4015@126.com (X.Z.); liuyn88@hotmail.com (Y.L.); fangxibi@jlu.edu.cn (X.F.)

**Keywords:** DLK1, mRNA, triglyceride, beef cattle, genetic polymorphism

## Abstract

**Simple Summary:**

Delta-like non-canonical Notch ligand 1 (DLK1) is a candidate gene associated with lipid metabolism. In order to verify the function of the *DLK1* gene on lipid metabolism in Chinese Simmental cattle, we identified the effect of *DLK1* on lipid metabolism in bovine fetal fibroblast cells (BFFs). At the same time, we also detected the relationship between single-nucleotide polymorphism sites (SNPs) of bovine *DLK1* with the economic traits and fatty acids composition in Chinese Simmental cattle, such as the carcass fat coverage rate, loin eye muscle area, and fat color score, etc. In the present study, we precisely constructed and transfected the overexpression and interference vectors of the *DLK1* gene in BFFs. the results showed that the overexpression of the *DLK1* gene could decrease the contents of intracellular triglycerides (TGs) while interference of the *DLK1* gene could increase the TGs contents. Gas chromatography analysis of the fatty acid composition showed that the contents of octanoate acid and γ-linolenate acid were regulated as the expression of DLK1 was altered. We found two SNPs and genes associated with these traits in Chinese Simmental by the restriction fragment length polymorphism (RPLF-PCR) detection method. In summary, we verified the effects of bovine DLK1 on fatty acid metabolism from the cellular level to population genetic polymorphism, which also paved the way for further studying of the effects of the *DLK1* gene on lipid metabolism in vivo.

**Abstract:**

In this study, we precisely constructed and transfected the overexpression and interference vectors in BFFs to evaluate the role of *DLK1* gene on lipid metabolism in vitro. The expression of of DLK1 in the mRNA and protein level tended to reduce, and TGs were significantly increased in the pGPU6-shDLK1 group compared to the control group (*p* < 0.05). The expression of DLK1 in the mRNA and protein level were increased in the pBI-CMV3-DLK1 group compared to the control group, and the TGs content showed a significant decrease in the pBI-CMV3-DLK1 group (*p* < 0.05). Meanwhile, we used the restriction fragment length polymorphism (RFLP-PCR) detection method to screen SNPs further to explore and analyze the relationship between the gene and the economic traits of 28-month-old Chinese Simmental and the fatty acids composition of cattle longissimus muscle. The result showed that two SNPs, IVS3 + 478 C > T and IVS3 + 609 T > G, were identified as being significantly associated with carcass and meat quality traits in Chinese Simmental, such as the carcass fat coverage rate, loin eye muscle area, and fat color score. In summary, our results indicated that *DLK1* can affect lipid metabolism in bovine and these two SNPs might be applied as genetic markers of meat quality traits for beef cattle breeding.

## 1. Introduction

Delta-like non-canonical Notch ligand 1(DLK1), a type 1 membrane glycoprotein, is a member of the epidermal growth factor-like family of homeotic proteins, which are typically involved in cell fate determination [1]. DLK1 functions as a negative regulator of adipocyte differentiation [2]. Moon et al. found that DLK1-null mice presented with obesity, increased serum lipid metabolites, skeletal malformation, and growth retardation [3]. These results indicated that DLK1 could play an essential role in the differentiation of adipocytes in mice. Furthermore, this may imply that the *DLK1* gene is involved in adipogenesis, which could further affect the meat quality in farm animals.

A study on the sheep *DLK1* gene indicated that the ectopic expression of DLK1 protein was associated with the inherited skeletal muscle hypertrophy in sheep [4]. The elevated mRNA level of *DLK1* was highly correlated to the late stages of myogenesis after the differentiation of chicken primary muscle cells [5]. Albrecht et al. demonstrated a higher expression level of the *DLK1* gene in less marbled muscle of Holstein steers than more marbled muscle of Japanese Black steers at slaughter, and the number of DLK1-positive cells was negatively associated with the fat content [6]. In pigs, the *DLK1* polymorphism of F2 offspring with various parental origins had correlations with meat quality traits (e.g., fat deposition and muscle mass) [7]. A synonymous mutation (C451T), located in exon 5 of *DLK1*, had no correlation with phenotypic traits in Qinchuan cattle [8]. Magee et al. conferred that there were significant associations between the bovine DLK1-DIO3 domain and carcass, fertility, and health traits in Holstein–Friesian dairy sires [9].

The results of the transcriptome analysis from our previous study showed that *DLK1* was a candidate gene associated with lipid metabolism in cattle. To evaluate the role of DLK1 on lipid metabolism, we constructed and transfected the overexpression and interference vectors in bovine fetal fibroblast cells (BFFs). The expression levels of DLK1 were determined by quantitative real-time polymerase chain reaction (qRT-PCR) and Western blot. The mRNA level of three genes related to lipid metabolism and the cellular content of triglycerides (TGs) and fatty acids were detected and compared between groups. Then, single-nucleotide polymorphism sites (SNPs) of the *DLK1* gene by the restriction fragment length polymorphism (RPLF-PCR) technique were identified and analyzed for their association with the economic traits of 28-month-old Chinese Simmental. These results provide fundamental data demonstrating the influence of the *DLK1* gene in bovine for further research, especially on lipid metabolism.

## 2. Materials and Methods

### 2.1. Experimental Materials

#### 2.1.1. Cell Line

BFFs were gifted from the Laboratory of Animal Genetics at Jilin University.

#### 2.1.2. Animals and Traits Analysis

In total, 237 28-month-old Chinese Simmental steers were provided by the cattle farm of Xilingol League, Inner Mongolian and were randomly selected from the offspring of a Simmental population of approximately 1000 female cows and 21 bulls. Animal experiments were performed in strict accordance with the guide for the care and use of laboratory animals by the Jilin University animal care and use committee (permit number: SYXK (Ji) 2012–0010/0011). All the samples collections were performed under aseptic conditions and all efforts were made to minimize animal damage. The measurement and analysis of the traits and fatty acid component in longissimus muscle samples of Chinese Simmentals were described in our previous study [10].

### 2.2. Construction of PGPU6-shDLK1 and PBI-CMV3-DLK1 Vectors

Short-hairpin RNA (shRNA) target sequences of bovine *DLK1* mRNA were screened and designed by BLOCK-iT™ RNAi Designer (Table 1). ShRNA with a 21-mer stem derived from the above sequence was cloned into pGPU6/GFP/Neo vector using *Bbs*I and *Bam*HI digestion. PBI-CMV3-DLK1 vector was generated by amplifying the bovine *DLK1* coding sequence and cloning the fragments to pBI-CMV3 plasmid (#631632, Clontech Laboratories, Mountain View, CA, USA) through *Bam*HI and *Cla*I digestion(#R0136V, #R0197V, New England Biolabs, MA, USA). Primers for the bovine *DLK1* coding sequence were designed by Primer 5.0 software (Table 1) and synthesized by GENWIZ^®^. All sequences were detected and validated in Shanghai Sangon biotech (Shanghai, China).

### 2.3. BFFs Culture and Transfection

Twenty-four hours before transfection, BFFs cultured with growth medium, including DMEM/F12 (Dulbecco’s Modified Eagle Medium: F12) (GIBCO, Grand Island, NY, USA) and 10% fetal bovine serum (PAA, Pasching, Austria), cells were incubated at 37 °C, 5%CO_2_, and 21%O_2_. When cells reached 80% confluency, pGPU6- shDLK1, PGPU6/GFP/Neo, pBI-CMV3, and pBI-CMV3-DLK1 plasmid were transfected to BFFs in accordance with the protocol of animal genetics, breeding, and reproduction experiment, College of Animal Science, Jilin University. For transfection, pGPU6/GFP/Neo-shRNA-DLK1 (symbolized by pGPU6-shDLK1) and pBI-CMV3-DLK1 plasmid DNA (1 μL) was diluted in 150μL of DMEM/F12, and mixed gently with fugene^®^ HD transfection reagent (PRE2311, Promega corporation, Madison, WI, USA). The mixture was mixed gently and incubated for 20 min at room temperature and was equally added to each well, and the cells were incubated at 37 °C in a CO_2_ incubator. Cells culture were exchanged with fresh growth medium 40 h after transfection. The expression of green fluorescence protein (GFP) in cells was observed under a fluorescence microscope (TE2000, Nikon, Tokyo, Japan).

### 2.4. Analysis of mRNA Levels of DLK1 and Lipid Metabolism-Related Genes

Total RNA was extracted from BFFs using an innuPREP RNA Mini Kit (Analytik Jena, Jena, Germany). RNA concentration was quantitated by applying a spectrophotometer (Thermo Scientific, Waltham, MA, USA). For detection of DLK1 and β-actin, cDNA was synthesized with 1µg of total RNA by using a PrimeScript^TM^RT reagent Kit (RR047A, Takara^®^ Biotechnology, Dalian, China) following the manufacturer’s instructions. Real-time PCRs were performed by using SYBR Green Real-Time PCR assays (TIANGEN^®^, Beijing, China). Primers were designed by Prmier6 (showed in Table 1) and synthesized by GENWIZ^®^. β-actin was used as an internal standard to normalize the mRNA expression level using the 2^−ΔΔCT^ method. The experiments were repeated three times.

### 2.5. Analysis of Protein Levels of DLK1

BFFs were homogenized in RIPA lysis buffer (Beyotime, Shanghai, China) mixed with protease inhibitor. The protein concentration was determined with a BCA protein assay (KGP902, KeyGEN BioTECH, Nanjing, China) following the manufacturer’s instructions. The protein concentration was determined with a BCA protein assay (KGP902, KeyGEN BioTECH, Nanjing, China) following the manufacturer’s instruction. Cell lysates were centrifuged at 10,000 *g* for 10 min at 4 °C, and the supernatants that contained the target protein were collected and transferred to polyvinylidene fluoride (PVDF) membranes (ISEQ00010/IPVH00010, Millipore, MA, USA). Then, the membranes were incubated with primary anti-DLK1 (1:750) and anti-β tubulin antibody (Bioss, Beijing, China) (1:10,000) overnight at 4 °C. This was followed by a 1.5-h incubation with secondary antibodies conjugated with horseradish peroxidase (HRP) (1:2000) (BioWorld, Bloomington, IN, USA). Bound antibodies were detected by highly sensitive chemiluminescence substrate (Super Signal West Femto Substrat, Thermo Scientific, Waltham, MA, USA), using a chemiluminescence imager (Tanon, Shanghai, China). The band intensity analysis of the blots was obtained using Image-Pro Plus software. The band intensities of DLK1 were normalized to the band intensities of β tubulin.

### 2.6. Determination of TGs Content in BFFs

In BFFs after overexpression or interference of the *DLK1* gene, the detection of TGs was performed with a tissue TGs assay kit (Applygen Technologies Inc., Beijing, China) according to the manufacturer’s protocol. The optical density of each sample was determined by a microplate reader (Yong Chuang SM600, Shanghai, China).

### 2.7. Determination of Fatty Acid Contents in BFFs

BFFs were mixed with Folch’s solution (CHCl3: CH3OH = 2:1, *v*/*v*) and the internal reference fatty acid (ginkgolic acid C13:0, Sigma 49962). The cell precipitates were filled with high-purity nitrogen and violently shocked for more than 1 h at 4 °C. Then, 0.88% NaCl was added, mixed at 4 °C, and centrifuged at 1500 r/min for 10 min. The lowest layer of liquid was transferred to a glass tube and blown with nitrogen at 55 °C until the chlorine evaporated completely. Samples were suspended with methyl-esterified mixed solution containing 35% BF3/methanol (14%), 45% methanol, and 20% hexane, filled with nitrogen at 90 °C for 1 h. Hexane and 0.88% NaCl were added to the mixed liquid. The final solution was placed in a liquid chromatography injection bottle for Gas Chromato graphic analysis.

### 2.8. The Polymorphism of Bovine DLK1 Gene

PCR primers were designed using Primer 5.0 software to amplify the bovine *DLK1* gene (Table 1). All target sequences were amplified in a 25-µL reaction volume, including 1 μL of bovine genomic DNA (50 ng/µL), 1 μL of forward and reverse primers (10 μM), 12.5 μL of 1× Taq PCR Green Mix (Vazyme, Nanjing, China), and 9.5 μL of nuclease-free H_2_O. Amplification of PCR started with an initial denaturation at 94 °C for 5 min, followed by 35 cycles with 94 °C for 30 s, annealing at 58 °C for 30 s, 72 °C for 40 s, and a final step at 72 °C for 10 min. PCR products were run in 1% agarose gel electrophoresis and visualized under UV light (Alpha imager ep, Alpha Innotech, San Leandro, Ca, USA). To determine the genotype of *DLK1* SNPs, PCR products were digested with *Ase*I or *Msp*I enzymes (#R0526V, #R0106V, New England Biolabs, Ipswich, MA, USA). The 10-μL digestion solution included 1.5 μL of buffer, 0.3 μL of Ase1 enzyme, 4 μL of PCR product, and 4.2 μL of nuclease-free H_2_O. The solution was digested at 37 °C for 2 h. Digested products were detected by 2.0% agarose gel electrophoresis and visualized under UV light to the genotype of DLK1.

### 2.9. Statistical Analysis

Some relevant data of SNPs on the *DLK1* gene were calculated according to the genotyping results, which were the genotypic frequency, Hardy–Weinberg test, linkage disequilibrium analysis, and polymorphism information content. The genotype frequencies and allele frequencies were calculated for the examined Chinese Simmental cross steers and were analyzed by the significance test. An analysis of the genotypic effects of the DLK1 gene was carried out using the GLM procedure of SPSS. The fixed model was:Y*_ijk_* = u + ys*_i_* + m*_j_* + e*_ijk_*,(1)
where Y*_ijk_* was the phenotypic observation of the kth individual from the Simmental breed of genotype *j* in the ith-year season, u was the population mean, ys*_i_* was the year effect of the ith-year season, m*_j_* was the genotype effect of the genotype *j*, and e*_ijk_* was the random residual effect corresponding to the observed value [11].

## 3. Results

### 3.1. Bovine DLK1 Gene Regions’ Location and Transcripts

The bovine *DLK1* gene is located on bovine chromosome 21: 65,626,343–65,634,828 and its coding sequences contain 1314 bps and the transcript has 5 exons. The open reading frame (ORF) encodes 308 amino acids. There are at least four variable splicing forms in bovine *DLK1* mRNA, which are triggered by the deletion of the last exon partial nucleotide (Figure 1).

### 3.2. The Expression of the DLK1 Gene in BFFs after Overexpression or Interference

According to the sequencing results, the inserted fragments in pGPU6-shDLK1 and pBI-CMV3-DLK1 plasmid were accurate (Figure 2). The green fluorescence of transfected cells was observed in each group under fluorescent microscopy after 24 h of transfection (Figure 3). This indicated that plasmids were successfully transfected into BFFs.

To investigate the effect of pGPU6-shDLK1 and pBI-CMV3-DLK1 vectors on bovine DLK1 expression, the mRNA and protein amount of DLK1 was analyzed by qRT-PCR and Western blot, respectively. The results showed that mRNA expression of *DLK1* in the pGPU6-shDLK1 group tended to reduce compared to that in the control group (Figure 4A). There was a trend in the pGPU6-shDLK1 group that the protein amount of DLK1 was lower than that in the control group (Figure 4C). Moreover, the expression level of *DLK1* mRNA in the pBI-CMV3-DLK1 group was significantly increased compared to the control group (*p* < 0.01, Figure 4A). However, it only tended to have a higher DLK1 protein level in the pBI-CMV3-DLK1 group compared to the control group (Figure 4C). The TGs content of BFFs after transfection with pGPU6-shDLK1 and pBI-CMV3-DLK1 was analyzed. The results suggested that the TGs content of the pGPU6-shDLK1 group was significantly increased compared to the control group (*p* < 0.05). Moreover, there was a significant decrease of the TGs content in the pBI-CMV3-DLK1 group (*p* < 0.05) (Figure 4B). These results of the TGs content further confirmed the inhibitory effect of DLK1 in BFFs.

### 3.3. The mRNA Expression of Genes Related to Lipid Metabolism in BFFs after Interference and Overexpression of Bovine DLK1

The mRNA expression level of the CCAAT/enhancer binding proteinα (*CEBPα)* gene in the pGPU6-shDLK1 group was extremely significantly increased compared with the control group (*p* < 0.01) (Figure 5A). However, the mRNA levels of peroxisome proliferator-activated receptor-γ (*PPAR γ)* and lipoprotein lipase (*LPL)* of the pGPU6-shDLK1 group were significantly lower than the control group (*p* < 0.05) (Figure 5A). Furthermore, the expression level of *LPL* mRNA in the pBI-CMV3-DLK1 group was significantly elevated compared to the control group (*p* < 0.05) (Figure 5B).

### 3.4. Determination of the Fatty Acid Content in BFFs after Interference and Overexpression of the DLK1 Gene

The result about the determination of the fatty acid content in BFFs revealed that three fatty acids were detected, including octanoate, linoleate, and γ-linolenate acid. The octanoate acid content in the interference group was significantly decreased compared with the control group (*p* < 0.05, Figure 6A), and the content of γ-linolenate acid in BFFs overexpressing DLK1 was significantly lower than the control (*p* < 0.05, Figure 6B).

### 3.5. Polymorphism of Bovine DLK1 Gene and Its Association with Economic Traits in Chinese Simmental Steers

#### 3.5.1. Two SNPs of the DLK1 Gene of Chinese Simmental Steers

According to the sequencing results of PCR products, there were two polymorphisms (IVS3 + 478 C > T and IVS3 + 609 T > G) located in the third intron of the *DLK1* gene in Chinese Simmental steers. The results of RFLP analysis indicated that three genotypes were detected at each SNP (Figure 7B,C).

#### 3.5.2. Genetic Diversity of the DLK1 Gene in Chinese Simmental Steers Population

The allele frequency and genotype frequency of two *DLK1* SNPs, IVS3 + 478 C > T and IVS3 + 609 T > G, in Chinese Simmental are presented in Table 2. The allele frequency at the IVS3 + 478 C > T locus was 0.865 for C base and 0.135 for T base. The highest and lowest genotype frequency of IVS3 + 478 C > T were CC (0.738) and TT (0.008), respectively. Furthermore, the T base frequency of IVS3 + 609 T > G (0.846) was higher than G base (0.135). The genotype frequency at the IVS3 + 609 T > G locus was 0.719 for TT, 0.234 for TG, and 0.028 for the GG genotype. The dominant genes of IVS3 + 478 C > T and IVS3 + 609 T > G were CC and TT, respectively. The analysis results show that the homozygous rate of the two SNP sites is higher than the heterozygous rate. In Chinese Simmental cattle, the information contents of IVS3 + 478 C > T and IVS3 + 609 T > G polymorphisms were 0.21 and 0.23, respectively, which belonged to a low polymorphic frequency (PIC value <0.25). The two SNP sites conform to Hardy–Weinberg equilibrium in their cattle populations. The SNP linkage analysis results are shown in Figure 8. According to the analysis results, a strong linkage relationship was found between IVS3 + 478 C > T and IVS3 + 609 T > G (D ‘= 0.959, LOD = 41.46, r^2^ = 0.732).

#### 3.5.3. Association of DLK1 Gene Polymorphisms with the Carcass and Meat Quality Traits in Chinese Simmental

Because the genotype counts for SNP IVS3 TT was only two Chinese Simmental steers, we compared the carcass and meat quality traits between the CC and TC genotype using the linear model. The association analysis results revealed that at the IVS3 + 478 C > T locus, the individual with the C allele homozygous, a higher value of the carcass length, marbling score, loin eye muscle area, and fat color score were observed compared to the individual with a heterozygote (*p* < 0.05). However, there were lower values of back fat thickness and carcass fat in steers that were C allele homozygous compared with CT genotype steers. Furthermore, the individuals with the TG genotype at IVS3 + 609 T > G had a higher kidney fat weight, carcass depth, back fat thickness, and fat coverage rate than steers with the TT genotype (*p* < 0.05). The values of the fat coverage rate in individuals of the TG genotype were also significantly higher than in that with the GG genotype (*p* < 0.05). The loin eye muscle area and fat color score of steers with the TG genotype were significantly lower than the GG genotype ones (*p* < 0.05) (Table 3).

As shown in Table 4, individuals with CT genotypes of IVS3 + 478 C > T locus had higher linoleic acid than steers with CC genotypes (*p* < 0.05). A higher content of linoleic acid and arachidic acid was observed in the TG genotype individuals of the IVS3 + 609 T > G locus compared to that in the TT genotype ones (*p* < 0.05) (Table 4).

## 4. Discussion

In the present study, the potential effect of bovine DLK1 on lipid metabolism was verified in BFFs through RNA interference and overexpression technology. However, there were no significant differences in the expression of *DLK1 RNA* interference expression in mRNA and protein levels. Only a knockdown trend was shown between the pGPU6-shDLK1 and negative control group. The reason may be that the target sequence of shRNA was located in the last exon (fifth exon) of the *DLK1* gene. The previous study of Fahrenkrug et al. showed three splice variants detected in the mixed tissue of bovine and the alternative selection of a splice donor located in the bovine fifth exon [12,13]. Hence, the shRNA in this experiment could only recognize part of the *DLK1* splice isoforms, which led to no significant decrease. Additionally, other factors might also lead to alteration of the *DLK1* expression, such as some functional genes involved in lipid or fatty acid metabolism, or regulated factors that can alter the expression of the *DLK1* gene. It may also be due to post-transcriptional mechanisms involved in the translation of mRNA to protein, which can delay the situation of protein accumulation [14].

Our study showed that knockdown and the increased expression of the *DLK1* gene can regulate other functional genes involved in the lipid metabolism pathway. For example, there was a significant decrease of *PPAR γ* and *LPL* mRNA in the interference group and increased mRNA level of *LPL* in the overexpression group. Both *CEBPα* and *PPARγ* were important transcriptional factors involved in lipogenesis and lipolysis in adipocytes [15,16]. *LPL* gene played a vital role in lipid metabolism. It can hydrolyze TGs core of lipoproteins so that the fatty acids and glycerol are generated for energy consumption and storage [17]. Moreover, the results of TGs content detection further supported this presumption. The content of TGs was significantly elevated in BFFs that interfered with shRNA but reduced significantly in the overexpression group compared to the respective control sample. It may suggest that lower bovine *DLK1* expression can promote TGs accumulation in cells whereas an increased expression level of bovine *DLK1* can decrease the cellular TGs content.

Furthermore, lower contents of octanoate acid and γ-linolenate acid were observed in the interference and overexpression group respectively. It has been reported that octanoate acid, a medium-chain fatty acid, increased glucose-stimulated insulin secretion from the rat pancreas [18]. High γ-linolenate acid was reported to be able to ease the symptoms of some chronic inflammation diseases [19]. Moreover, there were correlations between SNPs and fatty acid components, such as the content of linoleic acid and arachidonic acid. Phinney reviewed that various levels of arachidonic acid in the serum, liver, and muscle were associated with changes of the lipogenesis action [20]. As an essential fatty acid, linoleic acid can only be gained from the diet, and linoleic acid can be converted into γ-linolenic acid in the body [21]. The present study showed that the content of γ-linolenate acid was reduced in BFFs overexpressing DLK1 and a higher content of linoleic acid was observed in the TG genotype individuals of IVS3 + 609 T > G. These results could imply that the bovine DLK1 gene may influence the fatty acid content and expression of lipid metabolism-related genes in cells, thus affecting the lipid metabolism.

Magee et al. showed that there were correlations between the bovine DLK1-DIO3 domain and milk, carcass, and fertility traits in Holstein–Friesian cattle [9]. In our study, two SNPs (IVS3 + 478 C > T and IVS3 + 609 T > G) were identified in the third intron of the bovine *DLK1* gene. According to the association analysis results, these two SNPs were significantly correlated to carcass and meat quality traits, including back fat thickness, marbling, fat coverage rate, etc. As two SNPs of bovine *DLK1* demonstrated correlations to carcass and meat quality traits and fatty acid components in Chinese Simmental, they could be used as molecular markers for the cattle breeding process in the future.

In terms of the effect of DLK1 on lipid metabolism, -depth studies have proposed that the function of DLK1 is to shift the metabolic mode of the organism toward peripheral lipid oxidation and away from lipid storage, thus mediating important physiological adaptations associated with early life and with implications for metabolic disease resistance [22]. In obesity, a study not only identified two major adipokine clusters related to either body fat mass and inflammation or insulin sensitivity/hyperglycemia, and lipid metabolism (DLK1, SFRP5, and the like) but also revealed DLK1, Angiopoietin-like protein 6 (ANGPTL6), nicotinamide phosphoribosyl transferase (Nampt), and progranulin as the strongest adipokine correlates of type 2 diabetes (T2D) in obese individuals [23]. Therefore, it is very necessary to study the effect of the DLK1 gene on lipid metabolism in vivo or in vitro.

## 5. Conclusions

We accurately constructed pGPU6-shDLK1 and pBI-CMV3-DLK1 vectors. We detected and analyzed that the expression level of *DLK1* in BFFs after interference and overexpression can regulate the mRNA levels of the *CEBPα, PPARγ,* and *LPL* gene, which relate to lipid metabolism, and alter the content of octanoate acid, γ-linolenate acid, and TGs. Moreover, two SNPs of the *DLK1* gene were identified and shown to be associated with carcass and meat quality traits in Chinese Simmental steers. These results indicated that the bovine *DLK1* gene can affect the lipid metabolism, and two SNPs may be applied as molecular markers for beef cattle breeding selection in the future. Therefore, further studies need to be performed to reveal the mechanism of the *DLK1* gene on lipid metabolism and its influence on meat quality.

## Figures and Tables

**Figure 1 animals-10-00923-f001:**
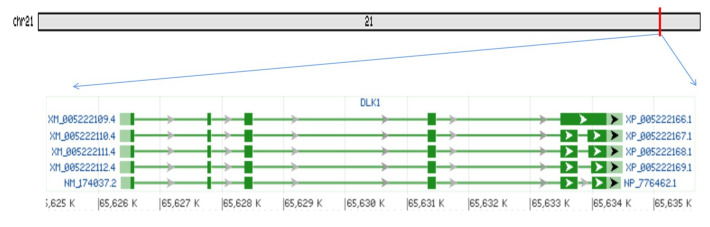
Five variable splicing forms of the bovine *DLK1* gene and all four variable splicings are triggered by deleting the last exon partial nucleotide.

**Figure 2 animals-10-00923-f002:**
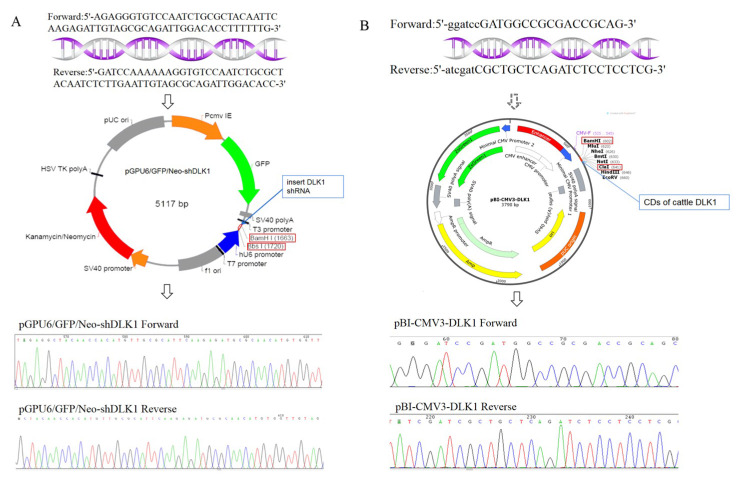
The *DLK1* gene interference vectors and overexpression vectors. Primer sequence of the RNA interference target sequence and interference vectors (pGPU6-shDLK1) (**A**), the primer sequence of coding sequences and overexpression vectors (pBI-CMV3-DLK1) (**B**).

**Figure 3 animals-10-00923-f003:**
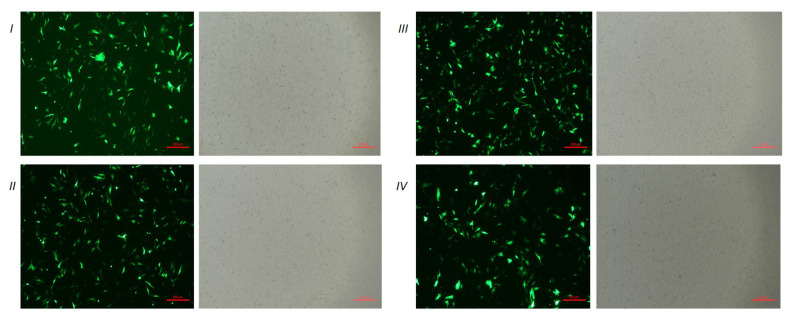
Expression of green fluorescence protein was observed in two vector groups under fluorescent microscopy. I: Cells transfected with pGPU6-shDLK1 vectors, II: Cells transfected with pGPU6/GFP/Neo vectors, III: Cells transfected with pBI-CMV3-DLK1, IV: Cells transfected with pBI-CMV3 vectors.

**Figure 4 animals-10-00923-f004:**
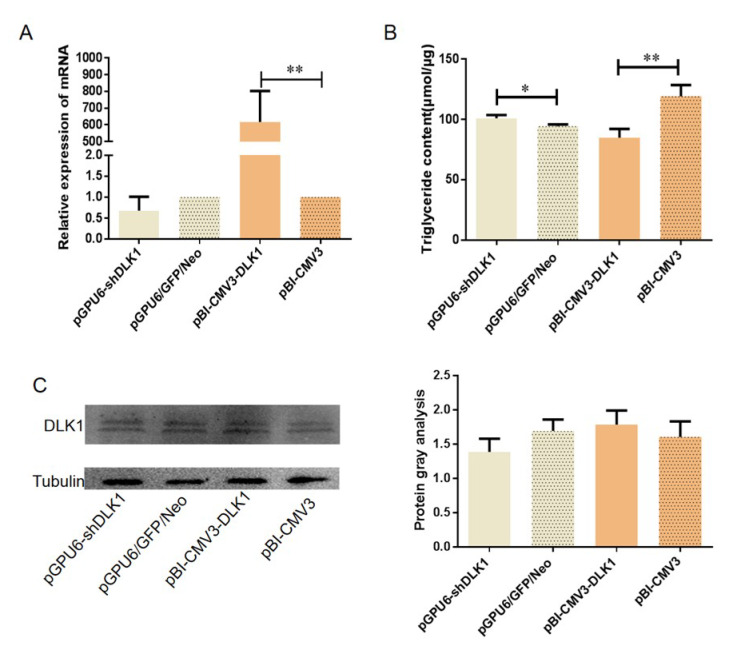
Effect of RNA interference and overexpression on mRNA, protein expression levels of *DLK1*, and Triglycerides (TGs) content. MRNA expression level of *DLK1* (**A**), TGs content in Bovine fetal fibroblast cells (BFFs) transfected with pGPU6-shDLK1 and pBI-CMV3-DLK1vectors (**B**), and protein expression levels of DLK1 (**C**). * means *p* value <0.05, ** means *p* value <0.01.

**Figure 5 animals-10-00923-f005:**
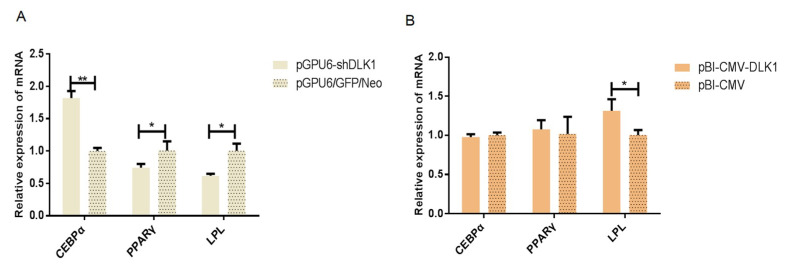
The mRNA expression of DLK1 and lipid metabolism-related genes in BFFs after interference (**A**) or overexpression of (**B**) the DLK1 gene. * means *p* value <0.05, ** means *p* value <0.01.

**Figure 6 animals-10-00923-f006:**
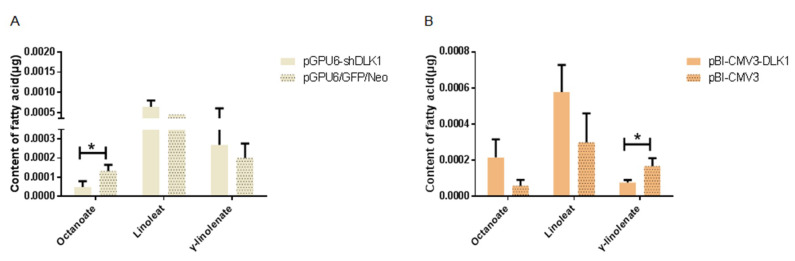
Comparison of the fatty acid content in BFFs after interference (**A**) or overexpression (**B**) of the bovine DLK1 gene. * means *p* value <0.05.

**Figure 7 animals-10-00923-f007:**
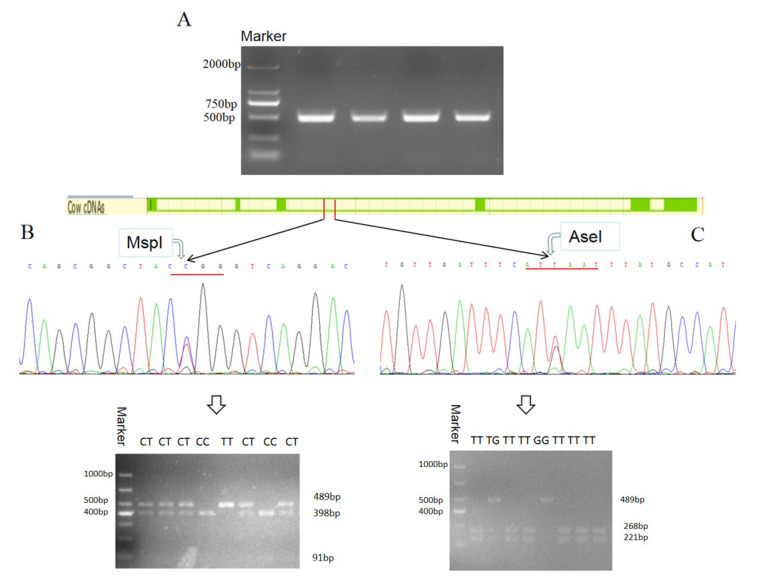
Detection and genotyping of Single-nucleotide polymorphism sites (SNPs) of the *DLK1* gene in Chinese Simmental steers. The DNA product was electrophoresed to obtain target fragment 489 bp (**A**). Both SNPs have three genotypes, one (IVS3 + 478 C > T) had the CC, CT, and TT genotype (**B**), and the other one (IVS3 + 609 T > G) had the TT, TG, and GG genotype (**C**).

**Figure 8 animals-10-00923-f008:**
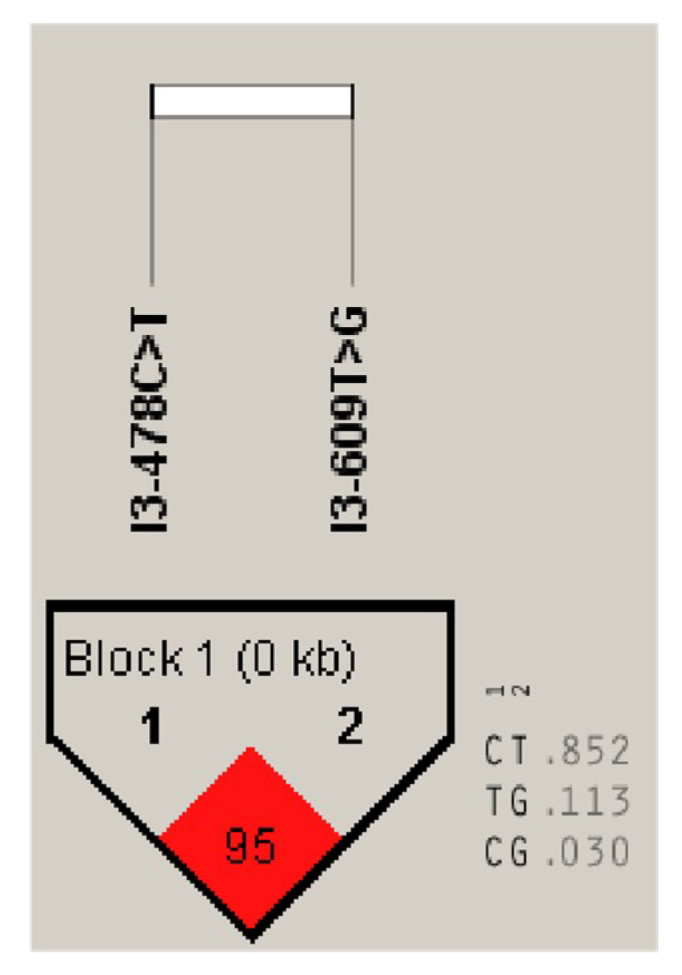
Linkage analysis of SNPs at the DLK1 locus in Chinese Simmental.

**Table 1 animals-10-00923-t001:** The primer sequences.

Primer	Forward Sequences (5′-3′)	Reverse Sequences (5′-3′)	Target Sequences
shRNA of bovine DLK1	AGAGGCTACAACCACATGTTGCGCATTCAAGAGATGCGCAACATGTGGTTGTAGCTTTTTTG	GATCCAAAAAAGCTACAACCACATGTTGCGCATCTCTTGAATGCGCAACATGTGGTTGTAGC	GCTACAACCACATGTTGCGCA
surpass coding sequence of bovine DLK1	TCCGCAACCAGAAGCCCA	GAGCGTAGCGTTCACCAGATTT	
coding sequence of bovine DLK1	GGATCCGATGGCCGCGACCGCAG	ATCGATCGCTGCTCAGATCTCCTCCTCG	
quantitative primer of bovine DLK1	CTTGCTCCTGCTGGCTTTCG	AGGTCACGCACTGGTCACAC	
quantitative primer of bovine LPL	CCGCAGACAGGATTACAG	GTGGTTGAAGTGACAGTTAG	
quantitative primer of bovine PPARγ	CCTTCACCACCGTTGACTTCTC	GATACAGGCTCCACTTTGATTGC	
quantitative primer of bovine C/EBPα	CCGTGGACAAGAACAAGCAAC	TGGTCAGCTCCAGCACCTTC	
Polymorphism primer of bovine DLK1	TCCACAGGTGAGGCTACTAAG	CTGTTCTCCTGACTTCCTAAG	
β-actin	AGAGCAAGAGAGGCATCC	TCGTTGTAGAAGGTGTGGT	

**Table 2 animals-10-00923-t002:** SNPs of the *DLK1* gene in Chinese Simmental.

SNP	IVS3 + 478 C > T(n = 237)		IVS3 + 609 T > G(n = 217)	
Location	Intron3		Intron3	
Gene frequency	C T	0.865 0.135	T G	0.846 0.154
Genotype frequency	CC CT TT	0.738 0.253 0.008	TT TG GG	0.719 0.234 0.047
PIC ^a^	0.21		0.23	
Ho ^b^	0.75		0.77	
He ^c^	0.25		0.23	

^a^ Polymorphic information content, ^b^ Ho: gene homozygote, ^c^ He: gene heterozygote.

**Table 3 animals-10-00923-t003:** Association of two SNPs of the bovine *DLK1* gene with economical traits in Chinese Simmental.

Economical Traits	Genotypes of IVS3 + 478 C > T		Genotypes of IVS3 + 609 T > G		
	CC	CT	TT	TG	GG
	Mean ± SD	Mean ± SD	Mean ± SD	Mean ± SD	Mean ± SD
LW (kg)	486.89 ± 56.00	481.99 ± 58.85	489.13 ± 58.37	486.72 ± 62.82	488.67 ± 68.85
CW (kg)	253.47 ± 35.00	249.99 ± 36.90	255.16 ± 35.89	252.48 ± 3 9.97	252.58 ± 46.58
KFW (kg)	4.44 ± 2.72 ^a^	5.07 ± 2.89 ^b^	4.53 ± 2.77 ^a^	4.96 ± 2.88 ^a^	3.46 ± 2.70 ^b^
CL (cm)	140.45 ± 9.01 ^a^	138.14 ± 6.53 ^b^	140.19 ± 8.96 ^a^	138.79±6.94 ^b^	145.83±4.96
TL (cm)	6.70 ± 0.87	6.93 ± 0.89	6.72 ± 0.88	6.92 ± 0.90	6.55 ± 1.03
CD (cm)	64.42 ± 3.37	64.45 ± 3.19	64.46 ± 3.32 ^a^	64.57 ± 3.28 ^a^	67.50 ± 1.87 ^b^
BFT (cm)	0.92 ± 0.64 ^a^	1.05 ± 0.65 ^b^	0.94 ± 0.65 ^a^	1.05 ± 0.63 ^a^	0.70 ± 0.61 ^b^
FCR (%)	46.64 ± 22.16 ^a^	51.37 ± 20.46 ^b^	47.50 ± 22.04 ^AB^	52.29 ± 21.13 ^A^	39.17 ± 23.54 ^B^
MBS	5.45 ± 0.67 ^a^	5.20 ± 0.80 ^b^	5.41 ± 0.67	5.36 ± 0.75	5.00 ± 1.10
LEA (cm^2^)	78.50 ± 12.99 ^a^	75.17 ± 10.32 ^b^	79.09 ± 13.24	74.62 ± 10.22	78.50 ± 12.15
FCS	2.87 ± 0.94 ^A^	2.50 ± 0.98 ^B^	2.83 ± 0.94 ^AB^	2.42 ± 0.96 ^B^	3.33 ± 0.52 ^A^

Note: LW: live weight, CW: Carcass weight, KFW: kidney fat weight, CL: carcass length, TL: thickness of loin, CD: carcass depth, BFT: back fat thickness, FCR: carcass fat coverage rate, MBS: marbling score (the score range of marbling is from Nos. 1 to 9), LEA: loin eye muscle area, FCS: fat color score (the score range of fat color is from Nos. 1 to 7). ^a, b^ means with different letters were significant difference (*p* < 0.05), ^A, B^ means with different letters were significant difference (*p* < 0.01). SD, standard deviation.

**Table 4 animals-10-00923-t004:** The correlation analysis of *DLK1* gene SNPs with the fatty acids composition in the longissimus muscle of Chinese Simmental cattle.

Types of Fatty Acids	Genotypes (IVS3 + 478 C > T)		Genotypes (IVS3 + 609 T > G)		
	CC	CT	TT	TG	GG
	Mean ± SD	Mean ± SD	Mean ± SD	Mean ± SD	Mean ± SD
Myristic acid (c14:0)	0.020 ± 0.019	0.028 ± 0.021	0.020 ± 0.017	0.027 ± 0.023	0.024 ± 0.009
Myristic oleic acid (c14:1)	0.002 ± 0.006	0.004 ± 0.004	0.002 ± 0.004	0.004 ± 0.004	0.001 ± 0.002
Hexadecanoic acid (c16:0)	0.263 ± 0.222	0.341 ± 0.200	0.251 ± 0.175	0.332 ± 0.223	0.277 ± 0.087
Palmitoleic acid (c16:1)	0.029 ± 0.038	0.035 ± 0.023	0.026 ± 0.020	0.033 ± 0.026	0.023 ± 0.013
Margaric acid (c17:0)	0.012 ± 0.008	0.015 ± 0.007	0.012 ± 0.007	0.014 ± 0.008	0.011 ± 0.003
Heptadecenoic acid (c17:1)	0.005 ± 0.008	0.007 ± 0.006	0.005 ± 0.006	0.006 ± 0.006	0.005 ± 0.006
Stearic acid (c18:0)	0.189±0.120	0.241 ± 0.118	0.184 ± 0.110	0.235 ± 0.133	0.219 ± 0.044
Oleic acid (c18:1n9c)	0.389 ± 0.469	0.424 ± 0.230	0.346 ± 0.236	0.416 ± 0.255	0.349 ± 0.113
Linoleic acid (c18:2n6c)	0.098 ± 0.028 ^b^	0.118 ± 0.050 ^a^	0.096 ± 0.026 ^b^	0.120 ± 0.053 ^a^	0.098 ± 0.023
α-Linolenic acid (c18:3n3)	0.005 ± 0.006	0.009 ± 0.012	0.004 ± 0.005	0.009 ± 0.013	0.009 ± 0.002
Arachidic acid (c20:0)	0.000 ± 0.001	0.001 ± 0.002	0.000 ± 0.001	0.001 ± 0.002	0.000 ± 0.001
Eicosenoic acid (c20:1)	0.001 ± 0.003	0.001 ± 0.001	0.000 ± 0.001	0.000 ± 0.001	0.000 ± 0.001
Dohono-γ-linolenic acid (c20:3n6)	0.010 ± 0.002	0.010 ± 0.004	0.010 ± 0.002	0.009 ± 0.004	0.009 ± 0.002
Arachidonic acid (c20:4n6)	0.048 ± 0.011	0.056 ± 0.019	0.048 ± 0.010 ^b^	0.055 ± 0.020 ^a^	0.043 ± 0.005

^a, b^ means with different letters show significant difference (*p* < 0.05). SD, standard deviation.

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
