# Peer review of "Analysis of the Bovine DLK1 Gene Polymorphism and Its Relation to Lipid Metabolism in Chinese Simmental"

_animals, 2020, doi:10.3390/ani10060923_

Round 1

Reviewer 1 Report

In my consideration, most of my suggestions has been followed, and this paper has been sufficienty improved by the authors. Presently shows the requirements to be accepted for publication in Animals. 

Author Response

Dear Editor and Reviewers,

On behalf of the authors of the submitted manuscript entitled “Analysis of the bovine DLK1 gene polymorphism and its elation to lipid metabolism in Chinese Simmental” (animals-815533), we thank you so much for all your kind reviews and suggestions. Indeed, they are all meaningful. We have studied the comments carefully and have made corrections which will hopefully meet with approval. The main corrections in the paper (marked in blue) and the responses to editor and reviewer’s comments are as follows:

Q1: The authors indicated that “237 Chinese Simmental steers of 28-months-old were provided by the cattle farm of Xilingol League, Inner Mongolian and were randomly selected from the offspring of a Simmental population of approximately 1000 female cows and 21 bulls”. As these animals are from 1000 cows and 21 bulls. The limited number of bulls in the population suggests that some of the 237 animals are half-sib/full-sib family although animals were randomly selected. Therefore, an animal model that accounts for population structure (or background gene effects) must be used to reduce false positive rates of the SNP associations. Please refer to the paper by Kennedy et al. 1992. Journal of Animal Science, 70:2000-2012.

Response : Thanks for your kind comments. There may indeed be some half-sibs or full-sibs in population. Because our study is not a genome wide association analysis, we only considered two loci for association test. We actually considered sir effect in our statistics analysis. However, for the studied loci we found the sir effect is not significant, and association with sir effect show similar results.

Q2: The sample size is very small for the SNP association. In Table 2, the genotype counts for SNP IVS3 TT was only 237 x 0.008 = 1.896, which is less than 2 animals. If the genotype effect is modelled as in the statistical model (line 168), the very small sample size within the TT genotype class will likely lead to a high false positive rate. Therefore, the SNP effects should be modelled as an allele substitution effect.

Response : Thank you for your professional suggestion. Because the genotype counts for SNP IVS3 TT was only 2 Chinese Simmental steers, we compared the carcass and meat quality traits between CC and TC genotype using linear model. The association analysis results revealed that “At IVS3 + 478 C>T locus,the individual with the C allele homozygous, higher value of carcass length, marbling score, loin eye muscle area, and fat color score were observed compared to the individual with heterozygote (p < 0.05). But there were lower values of back fat thickness and carcass fat in steers with C allele homozygous compared with CT genotype steers.”

Q3: The model description such as “u is the lowest square mean of the observed values, ysi is the effective value of the ith-year season, mj is the effective value of genotype j” sounds odd. What is the lowest square mean? Should it be just overall mean? What is the effective value?

Response : The model and description of this study are revised as follows:“where yijk was the phenotypic observation of the kth individual from the Simmental breed of genotype j in the ith-year season, u was the population mean, ysi was the year effect of the ith-year season, mj was the genotype effect of genotype j, eijk was the random residual effect corresponding to the observed value”.

We tried our best to improve the manuscript and made some changes in the manuscript.  These changes will not influence the content and framework of the paper.

We appreciate for Editors/Reviewers’ warm work earnestly, and hope that the correction will meet with approval.

Once again, thank you very much for your comments and suggestions.

Sincerely yours,

Runjun Yang

Reviewer 2 Report

The Authors addressed all my comments, hence i will recommend "acceptance" of this manuscript.

Author Response

Dear Editor and Reviewers,

On behalf of the authors of the submitted manuscript entitled “Analysis of the bovine DLK1 gene polymorphism and its elation to lipid metabolism in Chinese Simmental” (animals-815533), we thank you so much for all your kind reviews and suggestions. Indeed, they are all meaningful. We have studied the comments carefully and have made corrections which will hopefully meet with approval. The main corrections in the paper (marked in blue) and the responses to editor and reviewer’s comments are as follows:

Q1: The authors indicated that “237 Chinese Simmental steers of 28-months-old were provided by the cattle farm of Xilingol League, Inner Mongolian and were randomly selected from the offspring of a Simmental population of approximately 1000 female cows and 21 bulls”. As these animals are from 1000 cows and 21 bulls. The limited number of bulls in the population suggests that some of the 237 animals are half-sib/full-sib family although animals were randomly selected. Therefore, an animal model that accounts for population structure (or background gene effects) must be used to reduce false positive rates of the SNP associations. Please refer to the paper by Kennedy et al. 1992. Journal of Animal Science, 70:2000-2012.

Response : Thanks for your kind comments. There may indeed be some half-sibs or full-sibs in population. Because our study is not a genome wide association analysis, we only considered two loci for association test. We actually considered sir effect in our statistics analysis. However, for the studied loci we found the sir effect is not significant, and association with sir effect show similar results.

Q2: The sample size is very small for the SNP association. In Table 2, the genotype counts for SNP IVS3 TT was only 237 x 0.008 = 1.896, which is less than 2 animals. If the genotype effect is modelled as in the statistical model (line 168), the very small sample size within the TT genotype class will likely lead to a high false positive rate. Therefore, the SNP effects should be modelled as an allele substitution effect.

Response : Thank you for your professional suggestion. Because the genotype counts for SNP IVS3 TT was only 2 Chinese Simmental steers, we compared the carcass and meat quality traits between CC and TC genotype using linear model. The association analysis results revealed that “At IVS3 + 478 C>T locus,the individual with the C allele homozygous, higher value of carcass length, marbling score, loin eye muscle area, and fat color score were observed compared to the individual with heterozygote (p < 0.05). But there were lower values of back fat thickness and carcass fat in steers with C allele homozygous compared with CT genotype steers.”

Q3: The model description such as “u is the lowest square mean of the observed values, ysi is the effective value of the ith-year season, mj is the effective value of genotype j” sounds odd. What is the lowest square mean? Should it be just overall mean? What is the effective value?

Response : The model and description of this study are revised as follows:“where yijk was the phenotypic observation of the kth individual from the Simmental breed of genotype j in the ith-year season, u was the population mean, ysi was the year effect of the ith-year season, mj was the genotype effect of genotype j, eijk was the random residual effect corresponding to the observed value”.

We tried our best to improve the manuscript and made some changes in the manuscript.  These changes will not influence the content and framework of the paper.

We appreciate for Editors/Reviewers’ warm work earnestly, and hope that the correction will meet with approval.

Once again, thank you very much for your comments and suggestions.

Sincerely yours,

Runjun Yang

This manuscript is a resubmission of an earlier submission. The following is a list of the peer review reports and author responses from that submission.

Round 1

Reviewer 1 Report

In this study, the authors verified the role of DLK1 gene on lipid metabolism in Chinese Simmental cattle. They identified the effect of DLK1 on lipid metabolism in bovine fetal fibroblast cells (BFFs) and detected the relationship between single-nucleotide polymorphisms (SNPs) of bovine DLK1 and Chinese Simmental traits. Although, the experiment is precisely planned and nicely executed, however, my concerns are as under.

The author identified SNPs and their frequencies, however, information regarding genetic indices including HWE equilibrium, Polymorphism information content between these SNPs (PIC), Heterozygosity (He), and effective allele numbers information are missing. The authors also did not calculate the linkage disequilibrium (LD) among these SNPs.

The author also analyzed TG, however, they did not show the lipid droplets in the cells. I will suggest that author should differentiate the cells and show pictures of lipid droplets in cells transfected with both overexpressed and interference vectors.

Other minor comments

Line 73. Why BFFs cell line was selected for this study, please explain?

Line 92-98. How cells were transfected? Please explain in detail.

Line 112. Check it for spelling and grammar.

Line 134. What is total number of animals for genotyping?

Line 135-139. Check sign of degree (°).

How the phenotypic data of carcass traits were generated for association analysis?

Line 147-155. How about bull effect in the statistical analysis?

Line 159-162. Sentences are not clear; Figure 1 is also not clear.

Line 218-220 and so on the whole manuscript regarding genotyping/variants presentation. The authors must provide the proper nomenclature for the variants identified. The information can be found at http://www.hgvs.org/mutnomen/.

Para 262-273 are confusing needs rephrasing.

Author Response

Dear reviewer,

On behalf of the authors of the submitted “Analysis of the Bovine DLK1 Gene Polymorphism and its relation to Lipid Metabolism in 28-month-old Chinese Simmental”, we thank you so much for all your kind reviews and suggestions. Indeed, they are all meaningful. We have studied comments carefully and have made correction which we hope to meet with your approval. The main revision in the paper (highlight changes in red). 

Point 1: The author identified SNPs and their frequencies, however, information regarding genetic indices including HWE equilibrium, Polymorphism information content between these SNPs (PIC), Heterozygosity (He), and effective allele numbers information are missing. The authors also did not calculate the linkage disequilibrium (LD) among these SNPs.

Response 1: Thank you for suggestion. We have calculated the relevant data and added it to the article. The results were shown in line 247-254, Table 2 and Figure 8.

Point 2: The author also analyzed TG, however, they did not show the lipid droplets in the cells. I will suggest that author should differentiate the cells and show pictures of lipid droplets in cells transfected with both overexpressed and interference vectors.

Response 2: Thank you for valuable comments. We have done oil red staining of BFFs, but the changes of lipid droplets are not significant. Moreover, lipid droplets have uneven size were observed in BFFs as difficult as lipocyte cells. It is difficult to determine the change in lipid droplets to the same field of view after staining.

Point 3: Line 73. Why BFFs cell line was selected for this study, please explain?

Response 3: Thank you for your question. The DLK1 gene is selected from the transcriptome analysis of our laboratory which  is also a differentially expressed gene found in the longissimus dorsi muscle tissue of beef cattle breeds with different intramuscular fat contents. Therefore, we selected the BFFs to verified  the function of DLK1 gene. Furthermore, we do the same experiments in preadipocyte cells. However, we did not obtain the ideal results because of the lower transfection efficiency. Hence, we selected the BFFs to verify the effect of target gene on lipid metabolism and the regulated function on intramuscular fat contents.

Point 4: Line 92-98. How cells were transfected? Please explain in detail.

Response 4: Thank you for your question. “For transfection, pGPU6/GFP/Neo-shRNA-DLK1 (symbolized by pGPU6-shDLK1) and pBI-CMV3-DLK1 plasmid DNA (1μL) was diluted in 150μL DMEM/F12, and mixed gently with fugene® HD transfection reagent (Promega corporation, USA). The mixture was mixed gently and incubated for 20 min at room temperature and was equally added to each well, and the cells were incubated at 37℃ in a CO2 incubator. Cells culture were exchanged with fresh growth medium 40 h after transfection. Expression of green fluorescence protein (GFP) in cells was observed under a fluorescence microscope (NikonTE2000, Japan).”  We added to the article in line 102-108.

Point 5: Line 112. Check it for spelling and grammar.

Response 5: Thank you for suggestion. We have changed in the article. We  revised “The protein concentration was determined with a BCA protein assay (KeyGEN BioTECH, China) followed with manufacturer's instruction, quantified and lysates were separated on 5% concentrated gel, 12% polyacrylamide gelsseparation gel and transerred to PVDF membranses(Millipore, USA). ”  to “The protein concentration was determined with a BCA protein assay (KeyGEN BioTECH, China) following with manufacturer's instruction. Cell lysates were centrifuged at 10, 000 g for 10 min at 4℃, and the supernatants that contained the target protein were collected and transerred to PVDF membranses(Millipore, USA). ”  (Line 121-124)

Point 6: Line 134. What is total number of animals for genotyping?

Response 6: Thank you for suggestion. We added the number in the article.(Line 79)

Point 7: Line 135-139. Check sign of degree (°).

Response 7: Thank you for suggestion. We have checked and changed sign of degree (°) in the article.(Line 123-157)

Point 8: How the phenotypic data of carcass traits were generated for association analysis?

Response 8: We used the phenotypic data which were measured in our laboratory previous study described in  reference[10] which is called  after “Bovine lipid metabolism related gene GPAM: Molecular characterization, function identification, and association analysis with fat deposition traits”. An analysis of the genotypic effects of the DLK1 gene was carried out using the GLM procedure of SPSS.(Line 81-86,  Line166-167)

Point 9:Line 147-155. How about bull effect in the statistical analysis?

Response 9: 237 Chinese Simmental steers of 28-months-old were provided by the cattle farm of Xilingol League, Inner Mongolian and were randomly selected from the offspring of a Simmental population of approximately 1000 female cows and 21 bulls. Therefore, the bull effect is not considered.

Point 10:Line 159-162. Sentences are not clear; Figure 1 is also not clear.

Response 10: We have revised the sentence and have placed a clear picture.(Line 174-177)

Point 11:Line 218-220 and so on the whole manuscript regarding genotyping/variants presentation. The authors must provide the proper nomenclature for the variants identified. The information can be found at http://www.hgvs.org/mutnomen/.

Response 11: Thank you for suggestion. We have provided and changed the proper nomenclature in the entire article, namely made I3-478 C>T into IVS3 + 478 C>T, made I3-609 T>G into IVS3 + 609 T>G .

Point 12: Para 262-273 are confusing needs rephrasing.

Response 12: These paragraphs mainly analyzed the protein expression level of the cells transfected with the DLK1 plasmid did not change significantly due to the alternative splicing of DLK1. We revised the sentence in line 289-300.

Thanks again for the professional advice put forward by the reviewer, we made revisions and replies seriously. In addition, we revised the manuscript carefully in sections to improve English writing.

                                                                                                                                                                                                                 May 6, 2020

Reviewer 2 Report

This paper treats an interesting subject, the association between some SNPs of a bovine gen and several traits. Paper is promissory but shows several essential defects which must be improved prior to its consideration for its publication is Animals.

Redaction of the manuscript must be improved, and a native review of grammar is also needed.

Sample size must be showed, it is necessary to know if the associations detected are consistent. Presently the number of animals used in the research is not remarked.

Statistical analysis must be better described.

Discussion is short, there is not a balance between the large amount of results and the limited of the discussion.

Conclusions. – All this section must be reformulated. Presently it is a simple summary of the results. Conclusions must remark the meaning of the results for the science. Its projections for the sector, specifically its applicability.

There are some minor aspect to be improved:

 Line 3.- remove “28-month-old”, it is not necessary in the title

Line 17.- which traits?

Line 24.- remove “preliminary

Line 39.- Several keywords are repeated in the title. Substitute them

Lines 59-60.- remove “However, few studies were reported about the 60 function and mechanism of DLK1 gene in Chinese Simmental cattle”. This is not relevant for the readers.

Lines 75-82.- how many animals were sampled?

Lines 152-155.- Define the nature of the effects (random or fixed)

Figure 1, 2 and 7.- improve its quality. There is text not readable

Line 321.- remove “and”

For all this I must recommend a major review

Author Response

Dear reviewer,

   On behalf of the authors of the submitted “Analysis of the Bovine DLK1 Gene Polymorphism and its relation to Lipid Metabolism in 28-month-old Chinese Simmental”, we thank you so much for all your kind reviews and suggestions. Indeed, they are all meaningful. We have studied comments carefully and have made correction which we hope to meet with your approval. The main revision in the paper (highlight changes in red). 

Point 1: Redaction of the manuscript must be improved, and a native review of grammar is also needed.

Response 1: Thank you for suggestion. We have invited native speaker and colleagues who are relatively professional in English and revised the grammar of the full text again.

Point 2: Sample size must be showed, it is necessary to know if the associations detected are consistent. Presently the number of animals used in the research is not remarked.

Response 2: Thank you for professional advice. We added the number in the article.(Line 79)

Point 3: Statistical analysis must be better described.

Response 3: Thank you for your comment. We inspected and described statistical analysis in detail, like HWE equilibrium, Polymorphism information content between these SNPs (PIC), Heterozygosity (He), calculate the linkage disequilibrium (LD) among these SNPs.(Line162-171, Line 247-254, Table 2 and Figure 8. )

Point 4: Discussion is short, there is not a balance between the large amount of results and the limited of the discussion.

Response 4: Thank you for your comment. We have inspected and enhanced the discussion. (Line333-341)

Point 5:  Line 3.- remove “28-month-old”, it is not necessary in the title

Response 5: Thank you for your comment, we have removed “28-month-old”in the title. The title of the current article is “Analysis of the Bovine DLK1 Gene Polymorphism and its relation to Lipid Metabolism in Chinese Simmental”.(Line2-4)

Point 6:  Line 17.- which traits?

Response 6: “The economic traits of Chinese Simmental, and fatty acid’s composition of cattle longissimus muscle, like carcass fat coverage rate, loin eye muscle area and fat color score.” We have added the detected traits in Line 17-19.

Point 7:  Line 24.- remove “preliminary”

Response 7: Thank you for your suggestion, we removed “preliminary”. The current content of the sentence is “we verified the effects of bovine DLK1 on bovine fatty acid metabolism from the cellular level and population genetic polymorphism.”(Line 26)

Point 8:  Line 39.- Several keywords are repeated in the title. Substitute them

Response 8: Thank you for your suggestion. We substituted several keywords, namely “Keywords: DLK1; mRNA; triglyceride; beef cattle; genetic polymorphism”.(Line 43)

Point 9:  Lines 59-60.- remove “However, few studies were reported about the 60 function and mechanism of DLK1 gene in Chinese Simmental cattle”. This is not relevant for the readers.

Response 9: Thank you for your suggestion. We have removed them.

Point 10:  Lines 75-82.- how many animals were sampled?

Response 10: Thank you for your suggestion. We added the number in the article.(Line 79)

Point 11:  Lines 152-155.- Define the nature of the effects (random or fixed)

Response 11:  The genotype frequencies and allele frequencies were calculated for the examined Chinese Simmental-cross steers and were analyzed by the significance test. An analysis of the genotypic effects of the DLK1 gene was carried out using the GLM procedure of SPSS. The fixed model was: Yijk =u+ysi+mj+eijk . where Yijk is the observed value of kth individual from the Simmental breed, of genotype j, in the ith year-season; u is the least square means of the observed values; ysi is the effective value of the ith year-season(fixed effect); mj is the effective value of the genotype j(fixed effect); and eijk is the random residual effect corresponding to the observed value.(Line 166-171)

Point 12: Figure 1, 2 and 7.- improve its quality. There is text not readable

Response 12: Thank you for your suggestion. We have placed clear pictures.(Figure 1, 2 and 7.)

Point 13: Line 321.- remove “and”

Response 13: Thank you for your suggestion. We have  removed “and”.  The current content of the sentence is “Gene sequence search, primer design, vector construction, genotyping and data analysis, RNA extraction, original draft preparation: M. W., J.P. ”.(Line 353)

  Thanks again for the professional advice put forward by the reviewer, we made revisions and replies seriously. In addition, we revised the manuscript carefully in sections to improve English writing.

                                                                                              May 6, 2020
